# Alpha and Omicron SARS-CoV-2 Adaptation in an Upper Respiratory Tract Model

**DOI:** 10.3390/v15010013

**Published:** 2022-12-20

**Authors:** Gregory Mathez, Trestan Pillonel, Claire Bertelli, Valeria Cagno

**Affiliations:** Institute of Microbiology, Lausanne University Hospital, University of Lausanne, 1011 Lausanne, Switzerland

**Keywords:** SARS-CoV-2, virus adaptation, persistent infection

## Abstract

Severe acute respiratory syndrome coronavirus 2 (SARS-CoV-2) is currently causing an unprecedented pandemic. Although vaccines and antivirals are limiting the spread, SARS-CoV-2 is still under selective pressure in human and animal populations, as demonstrated by the emergence of variants of concern. To better understand the driving forces leading to new subtypes of SARS-CoV-2, we infected an ex vivo cell model of the human upper respiratory tract with Alpha and Omicron BA.1 variants for one month. Although viral RNA was detected during the entire course of the infection, infectious virus production decreased over time. Sequencing analysis did not show any adaptation in the spike protein, suggesting a key role for the adaptive immune response or adaptation to other anatomical sites for the evolution of SARS-CoV-2.

## 1. Introduction

Almost three years after the identification of a novel virus which causes pneumonia, severe acute respiratory syndrome coronavirus 2 (SARS-CoV-2), more than 600 million people have been infected [1,2,3]. Probably originating from bat coronaviruses, SARS-CoV-2 adapted to infect humans [1,4,5,6,7]. The first human infections likely occurred in a Wuhan market in China in December 2019 [1,8,9,10]. From the beginning of 2020, the virus spread worldwide, leading to huge efforts to better understand this new virus and to keep it under control.

One of the major issues in the spread of SARS-CoV-2 is the multiple waves of infections, linked to the continuous emergence of variants (such as Alpha, Beta, Delta, and Omicron) and subvariants. For instance, the Alpha variant appeared in September 2020 and possibly emerged from a chronic infection, although these patients were generally unlikely to spread the infection [11,12]. The emergence of the Omicron variant, in November 2021, is still under investigation. A passage to animals and back to humans, or the infection of a hidden population, are possibilities under scrutiny [13,14,15]. 

SARS-CoV-2 variants differ primarily due to mutations in the spike protein, resulting in either increased binding for angiotensin-converting enzyme 2 (ACE-2), the entry receptor, or immune evasion. For instance, the mutation D614G, which became dominant in February 2020 [4,16], confers enhanced binding to ACE-2 and increased transmissibility [17]. Multiple mutations (e.g., K417N, E484A, and N501Y) were shown to lead to decreased antibody neutralization, as demonstrated by the augmented risk of re-infection with the Omicron variant [4,14,17,18,19,20,21,22]. However, mutations can also occur in other structural and non-structural proteins, resulting in innate immune evasion and increased viral replication: R203K and G204R in the nucleocapsid enhanced viral fitness [23] and Δ500−532 in the non-structural protein 1 (NSP1) are associated with decrease in interferon type I [24]. 

Importantly, SARS-CoV-2 was also reported to adapt rapidly to cell lines and different hosts, with the appearance of some mutations currently present in circulating variants. For instance, in Vero E6 cell lines, the furin cleavage site is lost, whereas it has been shown to be re-acquired in cells expressing TMPRSS2 or in vivo [25,26]. Studies in primates and mice with isolates from 2020 observed the apparition of the H665Y mutation now present in Omicron subvariants [26,27]. Moreover, mouse-adapted SARS-CoV-2 harbored N501Y which was then associated with almost all variants of concern [27]. However, animal models present important differences with humans, such as difference in receptors, leading to a need for viral adaptation in mice, or the lack of severe symptoms in non-human primates.

Therefore, we focused on suitable human surrogate models to study SARS-CoV-2 evolution. Nevertheless, cell lines were not suitable because SARS-CoV-2 enters through different routes in cell lines in vitro and in vivo [17]. 

To overcome this limitation, we used MucilAir tissues, which are human-derived pseudostratified epithelia, derived from human donor biopsies, mimicking the human upper respiratory tract. These tissues are composed of secretory, ciliated, and basal cells [28,29], although we do not have detailed information about the presence of rare cell types present in the natural landscape of respiratory tract such as the pulmonary neuroendocrine, suprabasal, ionocyte, tuft, or even deuterosomal cells [30]. Unfortunately, these types of cells have similar gene expression to other cells, and their presence has not been evaluated in previous characterizations [30,31]. However, the main difference between the human respiratory tract and MucilAir is the lack of immune cells. In previous experiments with respiratory viruses, it was demonstrated that the tissues were unable to clear infections [32]. For instance, the addition of macrophages to the system has been shown to limit the propagation of the virus [33]. Although the adaptive immune system is not present, this respiratory tract model can still produce cytokines and chemokines [32,33,34]. In addition to immune cells and rare cell types previously cited, neurons, smooth muscles, and fibroblasts are missing compared with the natural human respiratory tract and could somehow play a role during viral infection [30]. However, to the best of our knowledge, this model is still one of the best in vitro options to study viral growth in the upper respiratory tract. 

In our study, we investigated whether the human respiratory tract per se can drive adaptations of SARS-CoV-2. We infected human-derived respiratory tissues with two variants of concern (Alpha and Omicron BA.1) for one month. During the infection, we monitored the level of RNA and the infectiousness of apically released viruses. At the end of this long infection, SARS-CoV-2 viruses were sequenced to reveal any adaptation. 

## 2. Materials and Methods

### 2.1. Cells and Viruses

Vero E6 given by Prof. Gary Kobinger originally derived from ATCC CRL-1586 were maintained in Dulbecco’s modified Eagle medium (DMEM), high glucose, and Glutamax (Thermo Fisher Scientific, Waltham, MA, USA) supplemented by 10% fetal bovine serum (FBS) (Pan Biotech, Aidenbach, Germany) and 1% penicillin/streptavidin (Sigma, St. Louis, MI, USA). MucilAir (human upper respiratory tissues) were purchased from Epithelix (Plan-les-Ouates, Switzerland). They were maintained according to the manufacturer’s protocol. The specimens of the two variants of concern were collected at the University Hospital of Lausanne (CHUV). Their consensus genome sequences and raw reads were submitted to GISAID (B.1.1.7: EPI_ISL_2359887, BA.1: EPI_ISL_7681695) and the ENA (B.1.1.7: ERR6094646, BA.1: ERR8526519).

### 2.2. Infection of MucilAir

PBS with calcium and magnesium (Thermo Fisher Scientific, Waltham, MA, USA) was added to the apical side of MucilAir before infection with SARS-CoV-2. Of the nasopharyngeal sample, 50 µL testing positive for SARS-CoV-2 (B.1.1.7 or BA.1) was used to infect the apical side of the respiratory tissue. After infection, MucilAir was washed with MucilAir medium. Respiratory tissues were maintained at 33 °C. In the first week of infection, SARS-CoV-2 viruses apically released were collected every day. The following week, until the 25 days post-infection, they were collected twice per week. For the collection, 200 µL MucilAir medium was added on the apical side for 20 min at 33 °C and then stored at −80 °C for further analysis. The basal medium was changed twice per week. Viral RNAs were quantified by RT-qPCR and plaque assay of the apical wash were performed on Vero E6 (see below). On the last day of infection, MucilAir was fixed with 4% formaldehyde. The last apical washes collected were sequenced by next-generation sequencing.

### 2.3. Infection of Vero E6

Infection of Vero E6 with several passages was performed as described previously [35]. Briefly, Vero E6 cells (3.5 × 10^5^ cells per well) were seeded on a 6-well plate. Cells were infected with B.1.1.7 SARS-CoV-2 (MOI 0.01) for 1 h at 37 °C. The inoculum was removed, and DMEM with 2.5% FBS was added. Three days post-infection, viruses were collected and titred by plaque assay. For the following passage, viruses collected in the previous passage were used for the infection (MOI 0.01). After 11 passages, viruses were sequenced by next-generation sequencing.

### 2.4. Plaque Assay

Plaque assays on Vero E6 cells were performed as described by Mathez and Cagno, 2021 [36]. Briefly, Vero E6 (10^4^ cells per well) was seeded in a 24-well plate. The apical washes of MucilAir were serially diluted and used to infect Vero E6 cells. After one hour of incubation at 37 °C, an overlay of 0.6% avicel gp3515 (SelectChemie, DuPont, Zurich, Switzerland) in 2.5% FBS DMEM was added. Three days post-infection, cells were fixed with 4% formaldehyde and crystal violet. Plaques were counted manually.

### 2.5. Immunofluorescence

MucilAir tissues were stained as described previously [32,34], with minor modifications. Tissues were blocked with 5% FBS and Tween 0.5% for 30 min at room temperature. J2 (1:500) (Nordic-MUbio, BC Susteren, The Netherlands) and β-tubulin IV (1:250) (Abcam, Cambridge, United Kingdom) were added for 1 h at 37 °C. After three washes with PBS Tween 0.05%, tissues were incubated with 1:2000 Alexa Fluor 488 anti-mouse and Alexa Fluor 594 anti-rabbit (Thermo Fisher Scientific, Waltham, MA, USA) for 1 h at 37 °C. Then, 4,6′-diamidino-2-phenylindole (DAPI) (0.5 µg/mL) was added for 5 min. Tissues were mounted on a slide with Fluoromount g (Thermo Fisher Scientific, Waltham, MA, USA). A Zeiss LSM-900 confocal microscope was used to visualize the tissue with a Plan-Apochromat 63×/1.40 objective. 

### 2.6. RT-qPCR and Next-Generation Sequencing

Supernatants were lysed with TRK lysis and RNA was extracted with the E.Z.N.A Total RNA kit (Omega Biotech, Norcross, GA, USA). RT-qPCRs of SARS-CoV-2 were performed with TaqPath 1-Step RT-qPCR (Thermo Fisher Scientific, Waltham, MA, USA) with primers and probe targeting the E gene (Fwd: 5′-ACAGGTACGTTAATAGTTAATAGCGT-3′, Rev: 5′-ATATTGCAGCAGTACGCACACA-3′, probe: 5′-ACACTAGCCATCCTTACTGCGCTTCG-3′) [37]. Next-generation sequencing was performed as described previously [35,38]. The consensus sequences represent mutations observed in more than 70% of the reads. Due to a technical issue, BA.1 sequencing was not performed between 22,916 and 23,012 (according to the reference sequence of Wuhan).

### 2.7. Statistics and Analysis

Experiments were performed with two or three MucilAir samples for each variant and in duplicate for in vitro experiments. Results are shown as the mean and SEM. The area under the curve analysis followed by an unpaired *t*-test was performed to compare curves. GraphPad Prism version 9.1 (San Diego, CA, USA) and Geneious Prime (San Diego, CA, USA) version 2022.01 software were used for the analyses.

## 3. Results and Discussion

Viruses, including SARS-CoV-2, continuously adapt themselves to their environment, including new hosting organisms, cells, and their surroundings. To observe the respiratory tissue adaptations of SARS-CoV-2, we infected human-derived upper respiratory tract tissue with two nasopharyngeal samples that tested positive for SARS-CoV-2 for one month. Samples were never grown on cells or in tissues and were fully sequenced before experiments.

The first phase of the infection showed an increase in viral RNA released that peaked on the third day of infection (Figure 1A). The Alpha variant (B.1.1.7) had a higher fold change (3.58 log increase) during the same period compared with Omicron (BA.1) (1.12 log increase). After this initial phase of infection, viral RNA slightly decreased and stabilized. After 25 days of infection, 9.73 × 10^8^ RNA copies were detected for B.1.1.7 compared with 3.96 × 10^7^ for BA.1.

Omicron presented lower viral RNA copies and produced fewer infectious viruses than Alpha (Figure 1B), but the difference was not statistically significant. Previous results showed a better fitness of Omicron in the bronchi if compared with Delta and an opposite trend in the lung [39,40,41]. The results in nasal cell lines, however, were more ambiguous. In one study, Omicron was shown to have a faster replication but lower viral peak [42], whereas in another, higher replication with comparable viral peak has been observed [41]. In the comparison of infectivity, it is important to take into consideration the possible under estimation of Omicron due to its reduced plaque-forming ability [41,43].

The discrepancy between the high RNA level and the low level of infectious virus is in line with reports of human infections, which highlighted the presence of viral RNA in nasopharyngeal samples, although there was a lack of viral isolation one week after the first positive PCR test [44,45,46]. In our case, infectious viruses could be detected until the end of infection, but a drop (2.04 log and 0.64 log decrease for B.1.1.7 and BA.1, respectively) was observed in the same period.

After 25 days of infection, through immunostaining, we evidenced that both variants caused a reduction in the number of ciliated cells and damaged the tissues (Figure 1C). This finding is in line with previous reports showing that SARS-CoV-2 principally infects this type of cells [47,48]. We could identify, for both variants, cells with active viral replication (through double-stranded RNA staining), although in larger numbers for Omicron than for Alpha (Figure 1C). If combined with the infectivity and genome quantification, these results could suggest a higher replication per cell for Alpha compared with Omicron; however, no single-cell measurements of the two variants are available to further confirm this hypothesis.

We then performed next-generation sequencing on the virus collected 25 days post-infection in the apical wash to identify any potential adaptations to our ex vivo model and compared the consensus sequences with the initial sequence of the original clinical sample. Surprisingly, BA.1 did not harbor additional fixed mutations, whereas after B.1.1.7 infection, a single mutation, R724K, was observed in all the replicates (Figure 1D,E). No minority variants were observed for both BA.1 and BA.1.1.7.

The R724K mutation located in the NSP2 was previously reported to be characteristic of a Beta variant detected in Brazil [49]. Although the function of NSP2 is still unclear, it was associated with inhibition of the antiviral state of cells [50,51]. The mutation of arginine to lysine does not change the charge of the amino acid, with a probable minor impact on NSP2; however, further research should clarify the role of this mutation.

We cannot exclude that mutations arose at shorter time points but were not selected, or could arise with longer infections; however, the decreased amount of viruses released over time and the altered architecture of the tissues are not suitable to evaluate this possibility.

In contrast, when we infected Vero E6 cells with B.1.1.7, five mutations were present after several passages (Figure 1D,E): L1853F was predicted to affect the stability of NSP3 [52]; T95I was found in several variants including BA.1 and B.1.526 [53,54]; N149 is a possible site of N-glycosylation that might be associated with increased sensitivity to antibodies [55]; R685H results in a loss of furin cleavage site and is a mark of in vitro adaptation [25,56]. These results further confirm the possible adaptation to the host cell as a driver of evolution in the absence of adaptive immunity and emphasize that MucilAir models more faithfully mimic natural infection conditions than Vero E6.

In conclusion, in our study, we infected a representative model of the upper respiratory tract with two SARS-CoV-2 variants of concern (B.1.1.7 and BA.1) for one month. We highlighted that the viral RNA can be still detected but with a decrease in infectivity of viruses released over time, possibly linked to the loss of ciliated cells observed. Both variants did not show adaptations to the upper respiratory tract model. Therefore, in our experimental setting, the emergence of new variants was not due to the adaptation to the upper respiratory tract. Further work could be performed to assess the adaptation of clinical samples in different tissue models (e.g., brain, intestinal, etc.) or in the presence of suboptimal concentrations of immune sera to assess the role of other tissue tropism and adaptative immunity in SARS-CoV-2 evolution.

## Figures and Tables

**Figure 1 viruses-15-00013-f001:**
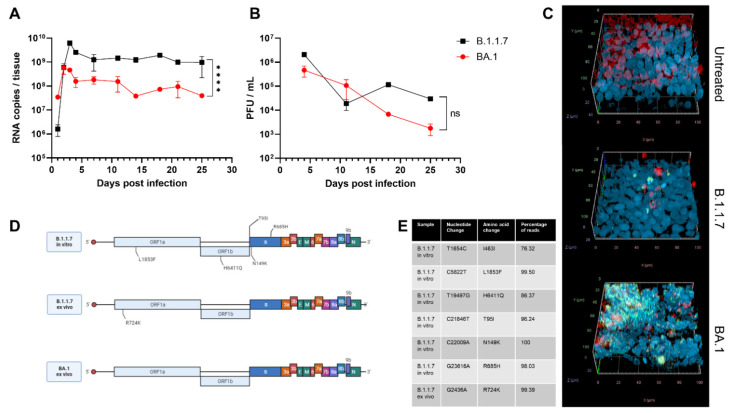
Evolution of SARS-CoV-2 variants ex vivo and in vitro after 25 days of infection. MucilAir tissues were infected with Alpha (B.1.1.7) or Omicron (BA.1) SARS-CoV-2. For 25 days, tissues were maintained at 33 °C and apical washes were performed to collect released SARS-CoV-2 viruses, quantified by RT-qPCR (**A**) and plaque assay (**B**). (**C**) Tissues were fixed and stained with J2 (dsRNA: green), β-tubulin IV (ciliated cells: red), and DAPI (nuclei: blue) at the end of infection. (**D**) Viruses were deep-sequenced at the end of infection ex vivo or after 11 passages in Vero E6. Mutations that were not present in the original sample are shown (created with biorender.com). (**E**) Nucleotide changes and percentage of reads present in the different samples. The results are the mean and SEM of two (B.1.1.7) or three (BA.1) MucilAir. *p*-value < 0.0001 (****).

## Data Availability

Raw data are available at 10.6084/m9.figshare.20596845 and sequenced data with accession number: ERS12330501, ERS12330502, ERS12330504, ERS12330505, ERS12330506, ERS12330507 and ERS12330508.

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
