# Peer review of "Alpha and Omicron SARS-CoV-2 Adaptation in an Upper Respiratory Tract Model"

_viruses, 2022, doi:10.3390/v15010013_

Round 1

Reviewer 1 Report

Reviewer #1: 

Coronavirus disease 2019 (COVID-19) has emerged as a new world pandemic, infecting millions of people with a substantial mortality. There is significant interest in understanding evolution SARS-CoV-2 in an upper respiratory tract model.

Recently publications show several evolution papers describe how several variants affect the pathogenicity, transmissibility and evasion immune. Therefore, this brief report could be help in understand this pandemic in the evolutionary context. 

In this manuscript, by Gregory et al titled "Alpha and Omicron SARS-CoV-2 adaptation in an upper respiratory tract model". The authors performed an analysis of the adaptive immune response or adaptation to other anatomical sites for the evolution of SARS-CoV-2.

There are several concerns that to be addressed.

This manuscript is well written and sites key findings in the field, therefore it will be helpful for investigators entering into coronavirus/COVID-19 research. The study would benefit the section on general aspects concern to SARS-CoV-2 evolution. Comments to improve the clarity of the manuscript are provided below.

Comments for the authors' consideration:

1.     The Introduction needs some work. I suggest expanding/citing more relevant, some papers about SARS-CoV-2 evolution in others model which have been published in the last two years. 

2.     The authors could show the deep sequencing data (% reads) of in a table close to figure D and also describe the changes at the nucleotide level.

Overall, results and discussion are well written.

Author Response

  1. The Introduction needs some work. I suggest expanding/citing more relevant, some papers about SARS-CoV-2 evolution in others model which have been published in the last two years. 

We thank the reviewer for his comment. To address this point, we added citations of work of SARS-CoV-2 evolution in primates and mice.

"Importantly SARS-CoV-2 was also reported to adapt rapidly to cell lines and different hosts, with appearance of some mutations currently present in circulating variants. For instance in Vero E6 cell lines the furin cleavage site is lost, while it has been shown to be re-acquired in cells expressing TMPRSS2 or in vivo [25, 26]. Studies in primates and mice with isolates from 2020 observed apparition of H665Y mutation now present in Omicron subvariants [26, 27]. Moreover, mouse adapted SARS-CoV-2 harbored N501Y that was associated with almost all variants of concern [27]. However, animal models present important differences with humans, for instance difference in receptors, leading to the need of viral adaptation in mice, or the lack of severe symptoms in non-human primates."

  1. The authors could show the deep sequencing data (% reads) of in a table close to figure D and also describe the changes at the nucleotide level.

We changed the figure 1 by adding a table with the nucleotide change and their corresponding amino acid change  with the percentage of reads from the deep sequencing result.

Overall, results and discussion are well written.

We thank the reviewer for the positive comment.

Reviewer 2 Report

In the present study, the authors investigated whether SARS-CoV-2 adaptation occurs in the upper respiratory model using an in vitro 3D human upper airway epithelium named MucilAir. In detail, at first they compared the amount of RNA and infectious virus production released by two VOC (B.1.1.7 and BA.1) over a one-month period and finally deep-sequenced to assess the emergence of any mutations at the end of the infection. Interestingly, the authors did not observe any particular adaptation of these two VOCs to the upper respiratory tract model.

Omicron presented lower viral RNA copies and produced lower number of infectious viruses than Alpha but as pointed out by the authors, some studies observed that ”Omicron is fitter than Alpha in the upper respiratory tract”.

1.     The paper cited by the authors (Hui et al., 2022) is not completely contradicting their results as they indicated since replication kinetics of several SARS-CoV-2 variants were tested in ex vivo cultures of human bronchi and lungs (lower respiratory epithelium).

2.     While the reviewer finds it very interesting that there is not a significant difference in the viral fitness between the two VOC, the authors should clarify their results, if they found as well a difference in the plaque-forming ability of each VOC and discuss their results with the literature (lines 152-155).

In Fig. 1C active replication for Omicron has been detected in more cells than for Alpha as suggested by immunostaining (dsRNA), while it presented lower RNA copies and infectious viruses than Alpha. This would suggest that fewer cells would have higher replication rate for Alpha than for Omicron. Is there a manner to quantify the replication rate per cell in order to clarify this point? Could the authors otherwise discuss this point?

While the reviewer understands the importance of such a model and that accumulation of mutations should be detected at the end of the experiment, is there any possibility that mutations occur during the same time period but are lost because not selected (due to the lack of a driving force as the immune system for example) and therefore it would be better to perform deep-sequencing more than once over the same time period?

Author Response

Omicron presented lower viral RNA copies and produced lower number of infectious viruses than Alpha but as pointed out by the authors, some studies observed that ”Omicron is fitter than Alpha in the upper respiratory tract”.

  1. The paper cited by the authors (Hui et al., 2022) is not completely contradicting their results as they indicated since replication kinetics of several SARS-CoV-2 variants were tested in ex vivo cultures of human bronchi and lungs (lower respiratory epithelium).

We thank the reviewer for the comment. We have now expanded the discussion of this result clarifying that Omicron was shown to be fitter in human bronchi, and including new references which evidence a better replication in the lungs, and  different results in nasal derived cell lines. Indeed the results in nasal cells are less clear and in part consistent with ours, with a more rapid replication and a lower peak as shown in Figure 1A.

"Previous results showed a better fitness of Omicron in the bronchi if compared to Delta and an opposite trend in the lung [40-42]. Results in nasal cell lines however are more ambiguous. In a study, Omicron showed to have a faster replication but lower viral peak [43] while in another, higher replication with comparable viral peak has been observed [42]."

  1. While the reviewer finds it very interesting that there is not a significant difference in the viral fitness between the two VOC, the authors should clarify their results, if they found as well a difference in the plaque-forming ability of each VOC and discuss their results with the literature (lines 152-155).

We observe a reduction of the ability of Omicron to make plaques. Reports in the literature support our observations. We clarified this point in the text.

"In the comparison of infectivity, it is important to take into consideration the possible under estimation of Omicron due to its reduced plaque forming ability [42, 44, 45]"

In Fig. 1C active replication for Omicron has been detected in more cells than for Alpha as suggested by immunostaining (dsRNA), while it presented lower RNA copies and infectious viruses than Alpha. This would suggest that fewer cells would have higher replication rate for Alpha than for Omicron. Is there a manner to quantify the replication rate per cell in order to clarify this point? Could the authors otherwise discuss this point?

The immunostaining was added to give a qualitative demonstration that we could identify cells with active viral replication. Unfortunately, we cannot extrapolate quantitative information from it. However, we included in the text this consideration and suggested future possible studies to evaluate the replication per cell for each variant, with single cell analysis.

While the reviewer understands the importance of such a model and that accumulation of mutations should be detected at the end of the experiment, is there any possibility that mutations occur during the same time period but are lost because not selected (due to the lack of a driving force as the immune system for example) and therefore it would be better to perform deep-sequencing more than once over the same time period?

We agree with the reviewer that other mutations could have been present during the course of the infection and being lost after 25 days post infection. We added a sentence to include this limitation. 

"We cannot exclude that mutations arose at shorter time points but were not selected, or could arise with longer infections, however the decreased amount of viruses released over time and the altered architecture of the tissues are not suitable to evaluate this possibility."

Considering our goal of finding mutations giving better fitness in the tissue model however, sequencing shorter time points will not provide additional information.